# Effects of Industry 4.0 on the Labor Markets of Iran and Japan

**Majid Ziaei Nafchi * 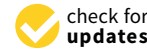 and Hana Mohelská**

Faculty of Informatics and Management, University of Hradec Kralove, Hradec Kralove 500 03, Czech Republic; hana.mohelska@uhk.cz

**\*** Correspondence: majid.ziaeinafchi@uhk.cz

**Abstract:** Industry 4.0 is the essence of the fourth Industrial revolution and is happening right now in manufacturing by using cyber-physical systems (CPS) to reach high levels of automation. Industry 4.0 is especially beneficial in highly developed countries in terms of competitive advantage, but causes unemployment because of high levels of automation. The aim of this paper is to find out if the impact of adopting Industry 4.0 on the labor markets of Iran and Japan would be the same, and to make analysis to find out whether this change is possible for Iran and Japan with their current infrastructures, economy, and policies. With the present situation of Iran in science, technology, and economy, it will be years before Iran could, or better say should, implement Industry 4.0. Japan is able to adopt Industry 4.0 much earlier than Iran and with less challenges ahead; this does not mean that the Japanese labor market would not be affected by this change but it means that those effects would not cause as many difficulties as they would for Iran.

**Keywords:** Industry 4.0; labor force; unemployment; Iran

## 1. Introduction

The first industrial revolution occurred around the end of 18th century by means of utilizing water and steam to power mechanical manufacturing facilities. The second industrial revolution happened in the early twentieth century when mass manufacturing was introduced by using electrical energy and counting on the division of labor. In the beginning of 1970s the third industrial revolution took place when productions were automated by the use of information technology and electronics.

Industry 4.0 (Industrie 4.0) which originated in Germany by using Cyber-physical systems (CPS), is the description for the 4th industrial revolution.

Preparing the very robust German manufacturing industry for the future by using integrated digitization, in an effort to be able to compete with the US, which is very dominant in the area of information and communication technologies (ICT), is the primary purpose of Industry 4.0 (Neugebauer et al. 2016).

The concept of Industry 4.0 has proved to be very attractive for other countries, and a number of those nations are considering enforcing Industry 4.0 in the near future, China is one of such countries. Nevertheless, implementing such advance technologies as Industry 4.0 is demanding and challenging especially in the case of developing countries. Iran could be considered as a great example of developing countries in this case.

The purpose of this paper is to find out whether it is possible for Iran and Japan to adopt Industry 4.0 with their current infrastructures, economy, and policies. Furthermore to investigate and analyze how far Iran is from implementing this kind of modern technologies like Industry 4.0 with the current

science and technology capabilities and policies, and to analyze the possible impacts that implementing Industry 4.0 could have on the labor market in developing countries in general and specifically in Iran, and to examine and compare such impacts with advanced countries.

## 2. Methodology and Literature Review

### 2.1. Methodology and Data

In this paper methods of literature review, data analysis and synthesis, methods of description, comparison, SWOT analysis, and induction and deduction reasoning approaches are carried out in an effort to examine and compare Iran as a developing country to Japan as an advanced country, and to answer the following questions:

- What is the effect of implementing Industry 4.0 on the labor market and particularly on the unemployment rate?
- What is the difference among labor markets in Iran and Japan?
- What are the impacts of implementing Industry 4.0 on the labor markets of Iran and Japan?
- What proves to be difficult for Iran and Japan in order to adopt Industry 4.0?
- How far is Iran from being capable to implement Industry 4.0?

Based on these questions the following hypothesis were established:

**Hypothesis 1 (H1).** *The impact of implementing Industry 4.0 is the same on the labor markets of Iran and Japan.*

**Hypothesis 2 (H2).** *The impact of implementing Industry 4.0 is not the same on the labor markets of Iran and Japan.*

The International Monitory fund (IMF), The World Bank (WB), and The Organization for Economic Co-operation and Development (OECD) didn't offer all the necessary data needed about Iran and Japan for the sake of comparison, or they simply didn't have such data that would be comparable. Therefore, the official webpage of the Central Intelligence Agency (CIA) was selected as a reliable source for collecting relevant information and economic indicators about the-mentioned countries. Figures used in this paper however, are from Statista and are based on data from IMF and WB.

The researcher uses triangulation—verification of the research conclusions deduced from multiple data obtained by different methods. In case of applying qualitative research strategy, it is possible to confront the research conclusions with the respondents themselves. Hendl (2005) understands triangulation as a combination of various methods, various researchers, various groups or people, various spatio-temporal and theoretical perspectives. Hendl (2005) and Berg (2012) distinguish three basic types of triangulation:

1. Data triangulation—combining various data sources.
2. Researchers' triangulation—deploying more interviewers and observers.
3. Methodology triangulation—combining within a single method or combining among more methods.

We used data triangulation and research questions arose in the course of research. This is also possible according to Hendl and Berg.

### 2.2. Industry 4.0

Kaeser and The World Is Changing (2018) considers the fourth industrial revolution as the best transformation ever recognized by human civilization, and believes that it could change and transform all human activities. The fourth industrial revolution could benefit more than ten billion people by the year 2050 if done correctly, but there is a high level of risk involved with it. The consequences of getting the fourth industrial revolution wrong are going to have severe impacts on the society.

Therefore, the fourth industrial revolution is about society as much as it is about business and technology. Meanwhile, the fourth industrial revolution is happening right now in manufacturing, called Industry 4.0 (Kaeser and The World Is Changing 2018).

All fields of the processes of the industrial production such as order management, manufacturing, research and development, delivery, utilization, commissioning, and also recycling of the products are altered and influenced by Industry 4.0. Neugebauer and his colleagues believe that the basis for new opportunities is the availability of useful and relevant information at the proper time and place. In order to have the relevant information available, every factor that is involved in the process (for example people, machines, and systems), has to be integrated into a system that is autonomously optimized and value adding which is dynamic and has the ability to self-organize (Neugebauer et al. 2016).

Integration and digitalization of simple technical-economic with associations to complex technical-economical networks, digitalization of proposals of products and services, and new market models are three mutually interconnected factors that are achieved by the use of Industry 4.0.

Currently these activities are interconnected by communication systems. Internet of Things (IoT), Internet of Services (IoS), and Internet of People (IoP) would be the most promising communications technologies in the Industry 4.0 environment in order to utilize data by being in charge of communications between communication entities (Zezulka et al. 2016).

Industry 4.0 allows manufacturers to make a digital twin of all the processes of their whole manufacturing in order to virtually design and simulate their state of the art products before actually producing that product physically. All the processes and tasks of manufacturing are optimized by the help of software to be performed be humans or machines. All the processes are tested in the virtual domain first, when they work properly in the virtual world then they are transferred into the physical world, which are the machines. The machines then have to report back to the virtual world to complete the loop. This smooth unification of the virtual and physical worlds is called cyber-physical systems (Kaeser and The World Is Changing 2018).

Neugebauer and his colleagues consider Cyber Physical Systems (CPS) as technical solutions that are connected by IoT, and the main objective of such systems is to decrease the distance between the physical and the digital domains. Therefore, smart solutions are required above the infrastructure, for instance the communication between individuals and physical systems (Neugebauer et al. 2016).

Optimizing value chains is the leading aim of Industry 4.0 and to serve this purpose, an autonomously controlled dynamic production is implemented. For this purpose, CPS are trusted to be very helpful instruments to grasp high stages of automation. When other technologies such as microcontrollers, actuators, sensors, and means of communication are combined with CPS then it would be possible to create smart factories (Kolberg and Zühlke 2015).

In order to achieve a more open and flexible value chains in manufacturing, Industry 4.0 has been developing CPS as well as dynamic production networks. The quality of information and the communication in the network environment are essential for such systems (Prause and Atari 2017).

Two models have been developed by Germany in order to digitize industrial production. These models are called Reference Architecture Model Industry 4.0 (RAMI 4.0), and Industry 4.0 Component Model, and they are currently considered to be the most important and essential models for the purposes of Industry 4.0 (Zezulka et al. 2016).

With the contemporary rate of progressions in science and technologies, the factory of the future would be like a very smart interacting organism that is able to learn, and would be very different from existing factories that operate on a set of processes, equipment and an accurate division of labor (Neugebauer et al. 2016).

Information technology unlocks the technologically innovative business processes and allows better sustainable business models in order to capture productivity and reducing costs. Adoption of technology oriented business process and information systems leads to the growth and sustainability in the organizations (Akhter 2017).

Technology transfer (TT) is considered to be an important factor for innovative business development and consequently for technological and economic development of a country. The time between when a scientist makes a discovery and when that discovery is adapted and used by businesses is a critical factor for success (Hilkevics and Hilkevics 2017).

### 2.3. Unemployment

The total number of workers in a country is called the labor force and it is classified as "employed" for those who have jobs, and "unemployed" for those who don't. Nevertheless, there are those which don't fit to the categories mentioned before (for example homemakers, retirees, and full-time students), these people are considered as "not in the labor force" and are not taken in to account when unemployment rate is calculated. The unemployment rate is the percentage of the people in the labor force that are not employed (Mankiw 2012).

Measuring the unemployment rate is simply the number of unemployed people divided by the total number of labor force multiplied by 100. Unemployment is one of apparent elements to determine a country's standard of living. People who lost their jobs or people who are not able to find jobs are not contributing to the country's economy. A country would have a higher GDP, growth rate, and higher standards of living when its workers are as fully employed as possible (Mankiw 2012).

### 2.4. Iran's Policies towards Science and Technology

According to the Iranian 20-year development plan which is called Vision 2025, Iran's government wishes to move to an economy that is based on knowledge rather than its current resource-based economy; and for this reason, instead of industries, Iran's policy-makers have concentrated their devotion to the country's human capital in order to create wealth. Therefore necessary steps were taken initially to increase the number of academics and university students, and furthermore to stimulate and inspire problem-solving and industrial research (UNESCO: Natural Sciences Sector 2017).

Mahdi, the author of "Evaluation of national science and technology policies in Iran" believes that Iran is relatively in a decent position considering basic sciences such as mathematics, physics, and chemistry in the world, and has substantial accomplishments in creating science. However, potentially there is a lot of room for producing science in Iran. Lots of underdevelopments are present and many improvements are required concerning innovation. In management of knowledge and research Iran is considered to be very week and consequently knowledge-based industries and services are not supported properly or not supported at all for their development. We cannot expect Iran to be able to develop and export knowledge-based products and services any time soon based on the current weak support that those industries receive (Mahdi 2015).

With the current circumstances and underdevelopments that exist in Iran, it appears very doubtful that Iran would have a prominent role in the world as for science and technology. In the direction of new and advanced technologies such as nanotech, nuclear, and aerospace, some initial steps have been taken by Iran; but then again as they are mentioned in the Iran's 20-year plan, vision 2025, no obvious outcomes of these developments have been influencing people's lives. The fact that the annual research budget in Iran is less than 1% of Iran's GDP could be one major reason. The annual budget has been slowly increasing but it is very doubtful that it would reach 4% of GDP by 2025, as it has been planned, and this is not enough to improve weaknesses and flaws that exist in fundamental research. Miracles cannot be expected with such low budgets in the field of research (Mahdi 2015).

Iran's educational system requires additional attention; Science and technology planners and policy-makers in Iran have been focusing their attention on the interaction of education with occupation for more than a decade now, then again no effective and important steps have been taken yet. Courses and educational degrees have little or no regard for Iran's overall scientific map and development of higher education in Iran has little regards for profession, production, and society. (Mahdi 2015) The research and development institutions have been expected to be more entrepreneurial

but very little is known about strategies and approaches to encourage and reach such objective (Khanagha et al. 2017).

Among other problems in Iran, we can mention the weak protection of intellectual properties. In order to have a strong protection of intellectual properties, there is a need for infrastructures and regulations in Iran more than ever, and this is not only to protect intellectual properties of other countries, but also domestic intellectual properties. The weak protection of intellectual properties has been limiting the existing policies and plans towards interaction and cooperation of other countries, and regional and global centers with Iran in the field of science and technology as it was highlighted in vision 2025. These policies were assessed as ineffective in the first place because of regional rivalry and political issues. Unfortunately in Iran, capturing, design, manufacturing knowledge, and transfer of technology are awfully low; and this is because there are low levels of cooperation and interaction of industrial countries with Iran due to political problems and limitations (Mahdi 2015).

The scientific and technical competencies of Iranian specialists who are living overseas could be very helpful but Iran's policies towards attracting them have not been successful yet. Unfortunately Iran at the current time doesn't have any policies or agendas to attract foreign specialists (Mahdi 2015).

The Iranian government has been censoring the Internet and is encouraging people to use the domestic networking platforms but people continue to use the western networking platforms. The main reason behind this is the technological advances, which the Iranian government couldn't keep up with; for example Secure Sockets Layer (SSL) and also Transport Layer Security (TLS) encryption (Small Media 2015).

## 3. Results

According to the 2016 estimates, Iran's Gross Domestic Product (GDP) was $1.459 trillion ($18,100 per capita) and the growth rate was estimated to be 4.5%; Iran's GDP by sectors was composed as follows: 9.1% from agriculture sector, 39.9% from industrial sector, and 51% from services sector. Some of the key agricultural products of Iran are wheat, rice, sugar beets, fruits, nuts, and cotton (Central Intelligence Agency 2017a).

The main Industrial products of Iran are considered to be petroleum, petrochemicals, gas, fertilizers, caustic soda, textiles, cement and other construction materials. Iran's government budget was around $65.87 billion in 2016 and it was mostly originated from oil and gas exports and the rest from taxes, and there was a 1.6% deficit (Central Intelligence Agency 2017a).

Japan's GDP was $5.238 trillion in 2016 estimations ($41,300 per capita) and the real growth rate of was estimated to be 1%; GDP was collected from the sectors as follows: 1.2% from agriculture sector, 27.7% from industrial sector, and 71.1% from services sector. Some of the major agricultural products are vegetables, tea, rice, fish, beef, pork, fruit, dairy products, flowers, potatoes/taros/yams, poultry, sugarcane, legumes, wheat and barley (Central Intelligence Agency 2017b).

The key industries of Japan are considered to be Motor vehicles, electronic equipment, machine tools, steel and nonferrous metals, ships, chemicals, textiles, processed foods. Japan's government budget was $1.696 trillion in 2016 and there was a 5% deficit (Central Intelligence Agency 2017b).

Population of Iran was estimated to be 82,801,633 inhabitants, the population growth rate was 1.18% and median age was 29.4 years in 2016. In Iran, 29.75 million people (36% of the total population) are considered to be in the labor force, and reports indicate that there is a shortage of skilled labor. According to earlier statistics in 2013, 16.3% of the labor force is occupied in agriculture sector, 35.1% in industry sector, and 48.6% in services sector. Iran has been repeatedly suffering from high unemployment and underemployment in the past decades and because of the lack in job opportunities; many young educated Iranians were forced leave the country in order to pursue jobs abroad. This has resulted in a significant brain drain (Central Intelligence Agency 2017a).

Population of Japan was 126,451,398 inhabitants according to the 2017 estimations, the population growth rate was −0.2% and the median age was 46.9 years. According to 2016 estimates 65.93 million

people (52% of total population) of Japan's population are considered to be in the labor force, and 2015 estimates show that 2.9% of the labor force was occupied in agriculture sector, 26.2% in industry sector, and 70.9% in services sector. The unemployment rate has been relatively low in Japan in the resent years (Central Intelligence Agency 2017b).

*3.1. Unemployment Rate*

Figure 1 illustrates the unemployment rate in the past few years as well as the forecasted unemployment rate for the following years to come. It is evident that Iran has been constantly suffering from a high unemployment rate and this high unemployment rate is expected to remain high in the years to come as the forecasted values show in this figure.

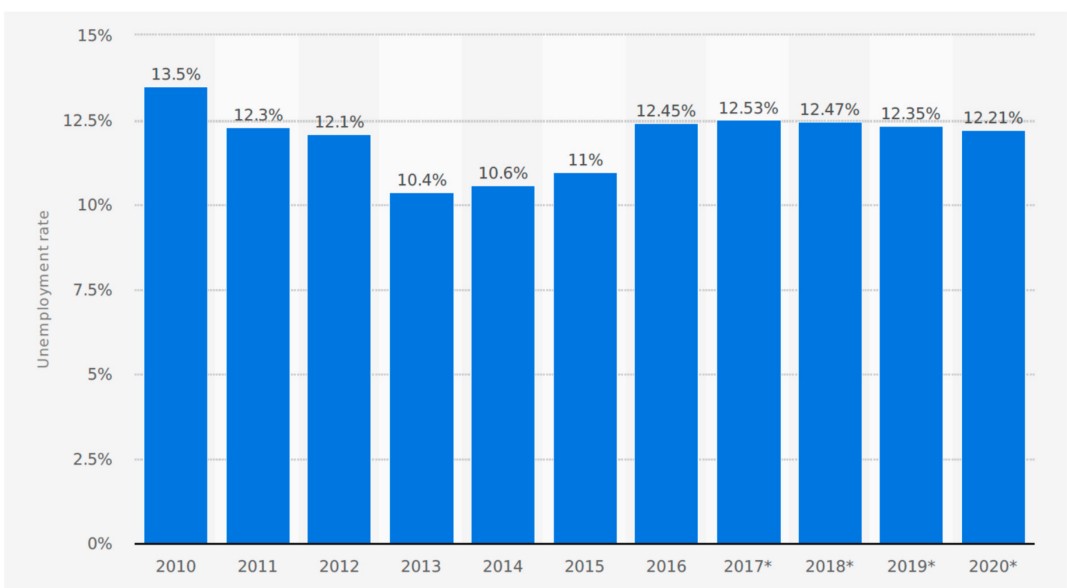

**Figure 1.** Unemployment rate 2010–2020 in Iran (Statista 2017d).

The unemployment rate of a country has many causes contributing to it. The youth unemployment (ages 15–24) is considered to be one of the main causes of the high total unemployment rate in Iran, and according to estimations in 2014 the youth unemployment was 24.0% in total, 21% male, and 42.8% female (Central Intelligence Agency 2017a).

One of the biggest socioeconomic concerns in Iran is the youth unemployment and the reasons causing it. Female youth unemployment is considerably higher than male youth unemployment rate, perhaps this could be explained by religious and cultural issues, limitations, and challenges that the female population is facing.

It is greatly improbable for Iran to have a low unemployment rate in the near future with the current high unemployment rate and the positive population growth rate, especially with other issues contributing to the unemployment rate, such as the phenomena that called the "brain drain" and also the lack of skilled workers.

Unlike Iran, not only Japan has a low unemployment rate but that low unemployment rate has been decreasing in the recent years as illustrated in Figure 2. In addition the unemployment rate will remain very low according to the forecasted values in the same figure.

The low unemployment rate in Japan shows that Japanese policy makers had successful policies towards keeping the unemployment rate as low as possible; nevertheless, it is hard to ignore the fact that Japan has a negative growth rate (Figure 4) and a much older population than Iran. According to the 2016 estimates, the median age was 46.9 (Central Intelligence Agency 2017b) in Japan, and 29.4 years (Central Intelligence Agency 2017a) in Iran; Thus, youth unemployment could be considered as a

significant contributing factor in Iran's total unemployment rate but we cannot say the same thing about Japan.

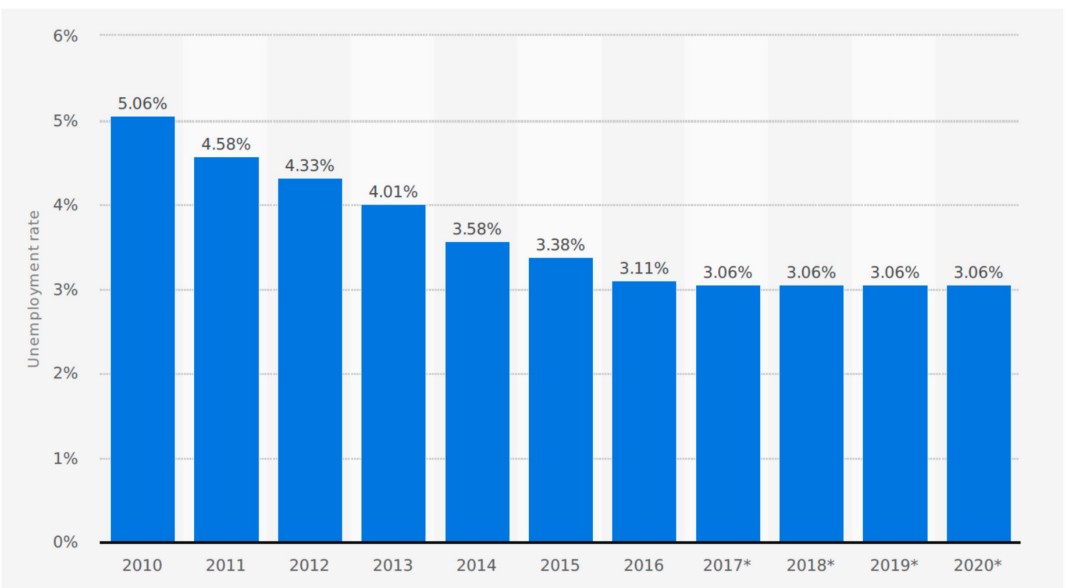

**Figure 2.** Unemployment rate from 2010 to 2020 in Japan (Statista 2017c).

*3.2. Population Growth*

As shown in the following figure (Figure 3) Iran's population has been growing since 2005 at a little bit over 1% every year. This positive growth rate means that in the near future there will be more people joining the labor market, which in turn will increase the number of people that are looking for jobs. To avoid this to become a problem or to add on the existing problems, the policy makers are to be more careful when making polices in order to create more jobs and to keep the unemployment rate as low as possible. Creating new job opportunities is very important and if not done according to the increasing demand for the jobs, the unemployment rate would continue to be high and it could possibly increase.

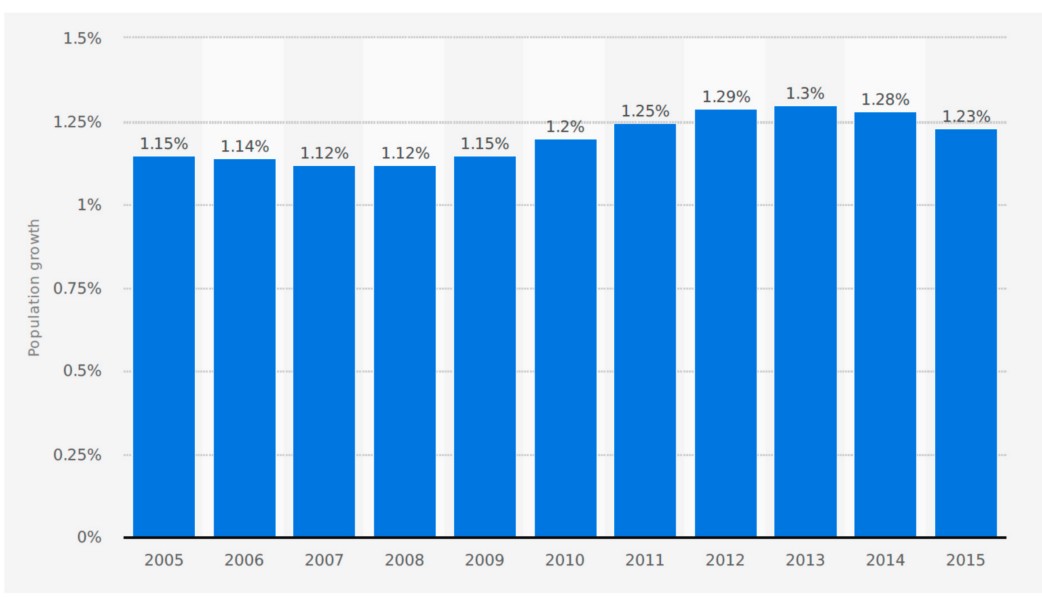

**Figure 3.** Population growth from 2005 to 2015 in Iran (Statista 2017b).

Figure 4 illustrates the population growth in Japan, as we can see the population is not growing at a steady rate. Japan had a positive population growth rate of 0.11% in 2007 and from 2011 forward Japan has had a negative population growth rate. The negative population growth in Japan's will affect the labor market; the labor market will eventually get older and smaller every year in the case that this negative growth will continue and this will decrease the unemployment rate slowly with the assumption that other factors would remain constant. These changes are very small but they are still important.

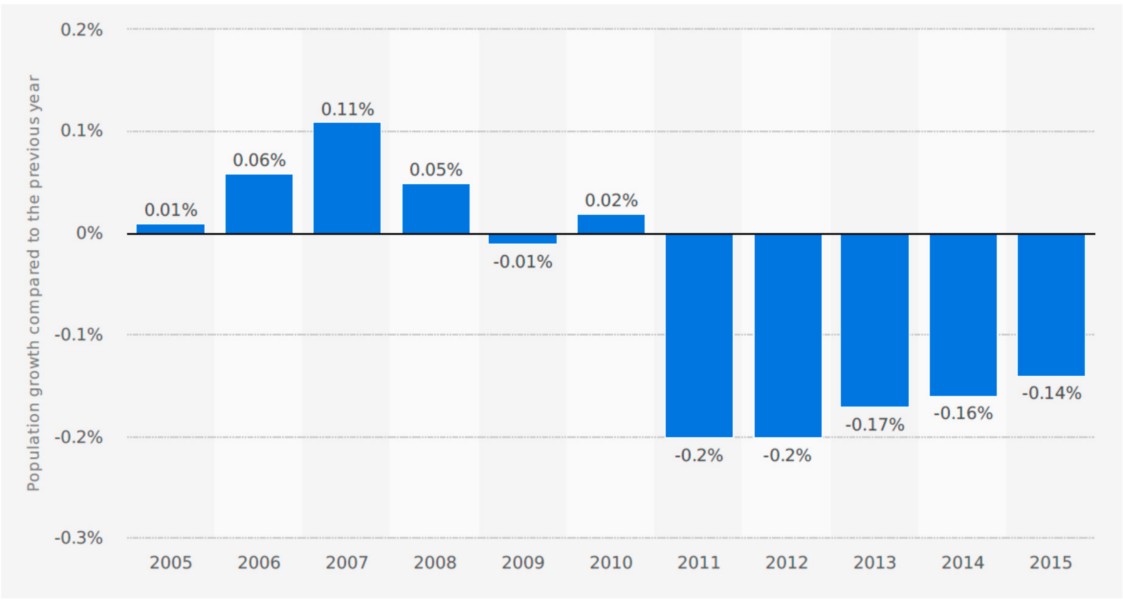

**Figure 4.** Population growth from 2005 to 2015 (compared to the previous year) in Japan (Statista 2017a).

## 3.3. SWOT Analysis

Authors carried out SWOT analysis for both Iran (Table 1) and Japan (Table 2), based on the information and analysis presented previously, to have a better perspective when comparing these countries on the subject of Industry 4.0.

**Table 1.** SWOT analysis for Iran.

| Strengths | Weaknesses |
|---|---|
| • A sizable and young labor force (about 36% of the population). <br> • Population growth rate is positive. <br> • Iran has a young population, this means most of the population is in the productive age (18 to 65 year old), which is a great potential and could be educated and trained for jobs that require skills and in order to support innovation and innovative processes. | • Skilled workers are needed. <br> • Unemployment rate is high. <br> • Weak protection of intellectual properties in Iran. <br> • Management of knowledge and research is very weak. <br> • Information is being censored by the government. <br> • Low government budget to support research and development. |
| **Opportunities** | **Threats** |
| • Opportunity to attract FDI. <br> • Opportunity to improve and develop infrastructures. <br> • Opportunity to lift international sanctions. <br> • Opportunity to increase competitive advantage. | • Risk of rising unemployment. <br> • International relations are weak. <br> • New international sanctions might be imposed. |

**Table 2.** SWOT analysis for Japan.

| Strengths | Weaknesses |
|---|---|
| • Japan has a large labor force (about 52% of the population). <br> • Advanced infrastructure. <br> • Protection of intellectual properties is strong in Japan. <br> • The unemployment rate is kept low. <br> • A high government budget to support research and development. <br> • Ongoing innovative thinking and innovative processes. | • Population growth rate is negative in Japan. <br> • The population of Japan is relatively old. This means that the number of people in the productive age (18 to 65 year old) is decreasing. |
| **Opportunities** | **Threats** |
| • Japan has very good international relations. <br> • Opportunity to help older workers. <br> • Opportunity to increase competitive advantage. | • Risk of rising unemployment. |

In the case of Japan SWOT analyses are mostly come as strengths and opportunities and therefore the strategy that is more appropriate for them is the offensive Strengths-Opportunity (SO) strategy and they have to utilize their strengths in order to take advantage of their opportunities, which is the using Industry 4.0. In other words Japan has to leverage its strengths to maximize the opportunities.

This is another story in the case of Iran; SWOT analyses came mostly in the form of Weaknesses and Threats. Therefore the strategy that is most appropriate with this is the defensive strategy Weaknesses-Threats (WT). Iran must overcome its weaknesses and try to minimize the threats.

## 4. Discussion

Kaeser and The World Is Changing (2018) the president and Chief Executive Officer of Siemens AG states that the fourth industrial revolution will eliminate and also create jobs as the other industrial revolutions but of course in a very larger scale. About 70% of the global trade comes from manufacturing therefore measures must be taken so people would benefit and at the same time be protected from the fourth industrial revolution (Kaeser and The World Is Changing 2018).

It appears that Iran's industrial sector and GDP would be affected more in comparison to Japan if they were to adopt Industry 4.0; as 35.1% of Iranian labor market is employed in the industrial sector making 39.9% of Iran's GDP, and 26.2% of Japanese labor market is employed in the industrial sector making 27.7% of Japan's GDP. However, these numbers are in percentage.

Iran's plans towards attracting Iranian specialists have been unsuccessful and regrettably there are no plans in order to attract foreign specialists to cover for the lack of skilled workers problem in Iran, therefore this problem is going to continue in the next few years to come and could have negative consequences on the unemployment rate. Necessary measures towards eliminating this problem have to be taken as soon as possible. Unlike Iran, Japan has no problems of this kind.

Iran's science and technology planners and policy-makers need to pay more attention to the education system; Improvements in the scientific map as well as some adjustments in courses and educational degrees are needed to be made by these planners and policy-makers in order to be in accordance with the occupation, production, and social needs.

Iran has not been successful to support the research programs as they are planned in vision 2025, especially that there was a deficit budget. In order to build the infrastructure and to improve research programs on the road to having a knowledge-economy, Foreign Direct Investment (FDI) could be a great solution that Iran can use. However, due to inconvenient international relations that Iran has, this is very challenging and less likely to happen.

Protection of intellectual property is very important and needs the policy-makers devotion to increase and improve the protection; better rules and regulations are needed to protect the domestic intellectual properties as well as those intellectual properties that belong to other countries.

Better protection of intellectual properties will help foreign countries feel safer and in return they would possibly be motivated to increase and expand their cooperation with Iran, and in different fields of science, technology, and research. In Japan intellectual properties are properly protected and they have lots of scientific cooperation with other nations.

For such a country like Iran that has many difficulties when it comes to the unemployment, adopting Industry 4.0 is not a good idea (assuming that it would be possible), since the essence of industrial revolutions and in particular the 4th one, is to decrease costs of production (good for Iran) by removing or reducing the human interaction, in other words automation, and this would heavily affect the labor force employed in the manufacturing and agricultural sectors. Keeping in mind that more than half of the Iran's labor force is employed in the before mentioned sectors, this could have a massive impact on the already existing unemployment problem, and a lot of people would lose their jobs. This doesn't mean that services sector wouldn't be affected like the other two sectors; but the impact of it would be far bigger on the agriculture and manufacturing sectors.

There are other obstacles that Iran has to overcome on the way to adapt Industry 4.0; there is a lack of basic infrastructure, education, research and technology essential for Industry 4.0 in Iran and on the other hand not enough high qualify programmers, IT and telecommunications specialists who are related with Industry 4.0 and Internet of Things (IoT).

Even with the assumption that Iran didn't have an unemployment problem, eliminating these obstacles and preparing requirements and necessary conditions for Industry 4.0 would take lots of years with Iran's existing policies and weak international relations.

The labor force in Japan, like any other country, would be affected if they adopted Industry 4.0. Nonetheless, it would be a bit different because they don't have unemployment problems like Iran in the first place. On the other hand, they have all the necessary science, technologies, and skilled workers that are required to support such a change; as well as a much higher budget and better international relations to help them complete this transition.

Kaeser and The World Is Changing (2018) emphasizes the following points as the way to insure that the fourth industrial revolution would be beneficial for everyone:

1. We have to learn from the past economic models and use them as the base of what is called inclusive society. One of these most successful models belongs to Germany and it's called social market economy and was developed by Alfred Müller-Armack. His vision was to have such an open society where the principal of the free market would be united with the fair distribution of well-being and prosperity. Joe Kaeser (2018) believes that paying attention to social responsibility and sustainability and business standards must be lifted up considerably in these areas; and this is the next step towards having inclusiveness. On the other hand, the business of business should not be only about the business as Milton Friedman's maxim and shareholder value should not be the benchmark on its own. Better options to use as a benchmark for a company's performance would be stakeholder value or social value. Essentially business of business is supposed to make value for the society, for example by taking care of the environment and climate, and educating employees and training workers (Kaeser and The World Is Changing 2018).

2. In order to enable workers to perform in a digital economy with the right qualifications and skills, some special attention is needed in the education and training programs from both businesses and the government alongside the fourth industrial revolution, as it is heavily dependent on Knowledge. (Kaeser and The World Is Changing 2018) There is the need to talk, think, and act today and not to leave the future of the industrial revolution to chance (Schwab 2018).

3. Digitalization has been proven to be very challenging and it has been changing industries, and digital technologies are changing business and social models as well. Therefore it is necessary to improve the ability to change and adapt and to promote innovation (Kaeser and The World Is Changing 2018).

4. All the difficult questions about the future of the Fourth Industrial Revolution must be addressed and not ignored. Questions like how to ensure the future of those people whose jobs will be eliminated, questions about how taxes should be imposed on software, and many more questions.

But it is important to move forward in order to take advantage of the opportunities and try to minimize the risks and try to answer those questions as we move ahead with the fourth industrial revolution. (Kaeser and The World Is Changing 2018) Klaus Schwab (the founder and executive chairman of the World Economic Forum Geneva) believes that the most pressing task governments have to deal with is to make room for new procedures to technology governance (Schwab 2018).

## 5. Conclusions

Industry 4.0 is particularly useful in very developed countries with regard to competitive advantage because wages are high in such countries. (Neugebauer et al. 2016) In developing countries like Iran, wages are low, there are relatively high levels of unemployment, and substantial developments are required in other basic but vital infrastructures.

CPR is used in Industry 4.0 in order to reach high levels of automation. Automation surges the unemployment rate with no regards for how well a country is developed. The mere difference is how large is the magnitude of the impact of this change on the economy of the country, and whether or not that country is able to cope with the consequences of such a change. It is evident in the SWOT analysis that implementing Industry 4.0 contains more challenges and would be more risky for Iran in comparison with Japan. Japan could benefit from the Offensive strategy of Strengths-Opportunity (SO) while the defensive strategy Weaknesses-Threats (WT) is more suitable for Iran.

Iran has a fairly young population, a positive growth rate, and a high unemployment rate that are less likely to be changed any time soon; so, adopting Industry 4.0 would impact the unemployment rate with the current positive growth rate and this could have overwhelming effects on the Iranian labor market.

In contrast, Japan has an older population, the unemployment rate is much lower comparing to Iran, has a negative growth rate, and has the required infrastructure and science and technology that is essential for implementing Industry 4.0. Hence, Japan is able to adopt Industry 4.0 much earlier than Iran and with less challenges ahead; this does not mean that the Japanese labor market would not be affected by this change but it means that those effects would not cause as many difficulties as they would for Iran.

Based on the analyses as Iran has a young population were the median age is about 17 years younger than Japan and because Iran has a positive population growth rate opposing the negative growth rate in Japan, it is safe to say that adopting Industry 4.0 will be more challenging for Iran in terms of unemployment and therefore the H1 is rejected and H2 is true: The impact of implementing Industry 4.0 is not the same on the labor markets of Iran and Japan.

Iran must make some serious decisions and take some major steps towards developing better international relations and cooperation, in order to be able to manage and increase research and development and above all in the field of knowledge management and applied sciences, and to develop the necessary infrastructures needed for Industry 4.0.

Iran has to protect intellectual properties, and this is something that needs the urgent attention of the policy-makers. Furthermore Iran needs to make better policies and programs to attract foreign specialists as well as Iranian specialist who live overseas in order. Brain drain is one more problem Iran is facing that must be taken care of immediately.

To encourage an increase in research and development in Iran, especially in the field of knowledge management and applied sciences, the government needs to increase the research budget. To do so, Foreign Direct Investment could be considered as a suitable solution.

With the present situation of Iran in science, technology, and economy, it will be years before Iran could, or better say should, implement Industry 4.0.

**Author Contributions:** Both authors have worked on the article together, their mental share is equivalent.

**Funding:** The paper was written with the support of the specific project 6/2018 grant "Determinants of Cognitive Processes Impacting the Work Performance" granted by the University of Hradec Králové, Czech Republic and thanks to help of student Tomáš Valenta.

**Conflicts of Interest:** The authors declare no conflicts of interest.

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
