# Peer review of "Effects of Industry 4.0 on the Labor Markets of Iran and Japan"

_economies, doi:10.3390/economies6030039_

Round 1

Reviewer 1 Report

 I suggest introducing to the text information about two more determines to reach high levels of automation in Industry 4.0 in Iran:

-  there are a various levels of technological advancement of the Internet in Iran and Japan and the performance of the Internet network (LTE, 4G) in Iran because this parameter also affects the revolution of Industry 4.0. Please read https://smallmedia.org.uk/media/articles/files/IIIP_DEC15.pdf

-  probably there are not enough high qualify programmers, IT specialists and telecommunications specialists in Iran who are related with Industry 4.0 and Internet of Things (IoT)

Author Response

- Suggestions were taken into consideration. 

- There is lack of skilled specialists in Iran.  

Reviewer 2 Report

Authors have to make more throurough critical review of contemporary literature before moving to their empirical part and interpretations.

Here are some papers, which, to my mind, should be included into the reviw of literature:

Akhter, F. 2017. Unlocking digital entrepreneurship through technical business process, Entrepreneurship and Sustainability Issues 5(1): 36-42. https://doi.org/10.9770/jesi.2017.5.1(3)

Prause, G.; Atari, S. 2017. On sustainable production networks for Industry 4.0, Entrepreneurship and Sustainability Issues 4(4): 421-431. https://doi.org/10.9770/jesi.2017.4.4(2)

Čirjevskis, A. 2017. Acquisition based dynamic capabilities and reinvention of business models: bridging two perspectives together, Entrepreneurship and Sustainability Issues4(4): 516-525. https://doi.org/10.9770/jesi.2017.4.4(9)

Hilkevics, S.; Hilkevics, A. 2017. The comparative analysis of technology transfer models, Entrepreneurship and Sustainability Issues 4(4): 540-558. https://doi.org/10.9770/jesi.2017.4.4(11)

Khanagha, A.; Dehkordi, A. M.; Zali, M. R.; Hejazi, S. R. 2017. Performance implications of entrepreneurial orientation at public research and technology institutions, Entrepreneurship and Sustainability Issues 4(4): 601-610. https://doi.org/10.9770/jesi.2017.4.4(15)

Author Response

The suggested papers were included in the article. 

Reviewer 3 Report

The article focused on the important topic that is under scope of the journal. Authors focused on the Industry 4.0 in Iran and in Japan.

The article has several deficiencies:

The citations are not used properly.

Only few references are included in WoS or Scopus.

A lot of text - whole paragraphs are taken only from sources without added value.

The situation of Iran is described in 2.4. Author can add 2.5 with situation in Japan.

Iran can be compared also with similar economy, that has / has not Industry 4.0 adopted.

SWOT analysis is not done as usual. It doesn´t contain typical evaluated criteria.

It looks like more as the conference paper than the journal paper.

Author Response

- More sources have been added to the article.

-Theoretical background has been improved.

- Methodology has ben improved.

- section 2.4 is addressing challenges and issues that Iran has, and there is no need for 2.5 since Japan is not facing those kind of issues.

-Iran could be compared with  a similar economy, that has / has not Industry 4.0 adopted but this could be done in future articles. 

- SWOT analysis were corrected.

Reviewer 4 Report

The topic of the article is very relevant throughout Europe and the world.

The aim of this paper is to find out how adopting Industry 4.0 could affect labor markets in Iran and Japan, and to make analysis to find out whether this change is possible for Iran and Japan with their current infrastructures, economy, and policies. 

This goal was not achieved.

The article's article focuses more on the situation in Iran than in Japan.

SWOT analysis is being discussed for Iran. It is in poor quality.

Contents of the article is not focused on the Industry 4.0, as indicated in the title of the article.

The article's contents are the basic macroeconomic indicators of Iran and Japan. No deeper static analysis is missing here.

The article's conclusions are general.

Literature review is only copied from scientific articles. I recommend doing a deeper literary research.

It is advisable to establish hypotheses.

Author Response

-Iran is actually meant to be the focus of this article

-SWOT analysis were improved 

- general improvements were made and more sources were added to the article

-Methodology has been updated 

Round 2

Reviewer 3 Report

Goal of the article presented in abstract was not achieved on the basis of the article.

It is advisable to establish hypotheses and to rework the goal of the article or the whole article.

The article in some parts compare Iran and Japan, in some parts is comparison missing.

SWOT analysis is still done in a poor quality. Japan has in opportunities involved increase of competitive advantage. Why it is not used also for Iran? There are more points for discussion in SWOT analysis.

I am missing deeper analysis between Industry 4.0 and labor market.

Some data are outdated, for example fig. 3 and fig. 4. For better evaluation will be better to have newer data and forecast for next periods.

Conclusions are general. Discussion is based mostly on citations. Citations are not always used properly.

Author use “forth industrial revolution”.

In this way paper looks like more as conference than journal paper.

Author Response

- Noted, the goal was corrected.

- Noted, Hypothesis were stablished.

- Only the more important points were compared.

- SWOT analysis were extended and improved.

- Figure 3 and 4 are only 2 years old and the content of them is not sensitive to time.

- Citations were checked and they are in accordance with the MPDI guidelines and template.

- Yes conclusions are general but chapter 3 includes the more detailed analyses.

- (Author use “forth industrial revolution”.) This comment is not clear.

- (In this way paper looks like more as conference than journal paper.) This comment is discouraging.  

Reviewer 4 Report

The aim of this paper is to find out how adopting Industry 4.0 could affect labor markets in Iran and Japan, and to make analysis to find out whether this change is possible for Iran and Japan with their current infrastructures, economy, and policies.

The goal of the paper has not been fulfilled.

The authors did not sufficiently implement the requirements in the first review. Only literature review is improved.

Missing hypotheses in the paper.

Swot analysis is not done well, missing conclusions and strategies.

All charts are from Statista. Where are the authors' own benefits and conclusions? There is no deeper analysis.

Author Response

- Noted, the goal was corrected.

- There were changes made to other parts of the manuscript (for example Methodology) not only the literature review.

- Noted, Hypothesis were stablished.

- Only the more important points were compared.

- SWOT analysis were extended and improved.

- Statista is a reliable source, there are no deep analyses but we hope that the analyses we have are sufficient for the purpose of this article.  

Round 3

Reviewer 3 Report

Goal of the article presented in abstract was achieved on the basis of the article. It could be set more narrowly.

The article in some parts compare Iran and Japan, in some parts is comparison missing.

Unfortunately, data in fig. 3 and 4 are outdated. More up-to-date are available.

Conclusions are general.

Author in some parts use “forth industrial revolution” instead „fourth“.

Author Response

- Noted, comments were taken into consideration

Reviewer 4 Report

Strengths:

Interesting topic

The discussion is useful.

Weaknesses:

Missing comments on swot analysis.

Poor conclusions

Hypothesis testing is missing

Lack of own analysis - prediction of the labor market development, etc. In the present form it is a summary of the facts that are known.

Author Response

- Noted,comments were taken into consideration 

Economies EISSN 2227-7099 Published by MDPI AG, Basel, Switzerland RSS E-Mail Table of Contents Alert
Back to Top